# Functional Characterization of Clinically-Relevant Rare Variants in *ABCG2* Identified in a Gout and Hyperuricemia Cohort

**DOI:** 10.3390/cells8040363

**Published:** 2019-04-18

**Authors:** Yu Toyoda, Andrea Mančíková, Vladimír Krylov, Keito Morimoto, Kateřina Pavelcová, Jana Bohatá, Karel Pavelka, Markéta Pavlíková, Hiroshi Suzuki, Hirotaka Matsuo, Tappei Takada, Blanka Stiburkova

**Affiliations:** 1Department of Pharmacy, The University of Tokyo Hospital, Tokyo 113-8655, Japan; ytoyoda-tky@umin.ac.jp (Y.T.); kmorimoto-tky@umin.ac.jp (K.M.); suzukihi-tky@umin.ac.jp (H.S.); tappei-tky@umin.ac.jp (T.T.); 2Department of Cell Biology, Faculty of Science, Charles University, 128 00 Prague 2, Czech Republic; andrea.mancikova@email.cz (A.M.); vladimir.krylov@natur.cuni.cz (V.K.); 3Institute of Rheumatology, 128 50 Prague 2, Czech Republic; pavelcova@revma.cz (K.P.); bohata@revma.cz (J.B.); pavelka@revma.cz (K.P.); 4Department of Probability and Mathematical Statistics, Faculty of Mathematics and Physics, Charles University, 121 16 Prague 2, Czech Republic; pavlikova@karlin.mff.cuni.cz; 5Department of Integrative Physiology and Bio-Nano Medicine, National Defense Medical College, Saitama 359-8513, Japan; hmatsuo@ndmc.ac.jp; 6Department of Pediatrics and Adolescent Medicine, First Faculty of Medicine, Charles University and General University Hospital, 121 08 Prague 2, Czech Republic

**Keywords:** ABCG2/BCRP, common disease, European cohort, exon sequence, functional study, gout susceptibility, heritability of serum uric acid, multiple rare variant, urate transporter, WGA

## Abstract

ATP-binding cassette subfamily G member 2 (ABCG2) is a physiologically important urate transporter. Accumulating evidence demonstrates that congenital dysfunction of ABCG2 is an important genetic risk factor in gout and hyperuricemia; recent studies suggest the clinical significance of both common and rare variants of ABCG2. However, the effects of rare variants of *ABCG2* on the risk of such diseases are not fully understood. Here, using a cohort of 250 Czech individuals of European descent (68 primary hyperuricemia patients and 182 primary gout patients), we examined exonic non-synonymous variants of *ABCG2*. Based on the results of direct sequencing and database information, we experimentally characterized nine rare variants of ABCG2: R147W (rs372192400), T153M (rs753759474), F373C (rs752626614), T421A (rs199854112), T434M (rs769734146), S476P (not annotated), S572R (rs200894058), D620N (rs34783571), and a three-base deletion K360del (rs750972998). Functional analyses of these rare variants revealed a deficiency in the plasma membrane localization of R147W and S572R, lower levels of cellular proteins of T153M and F373C, and null urate uptake function of T434M and S476P. Accordingly, we newly identified six rare variants of ABCG2 that showed lower or null function. Our findings contribute to deepening the understanding of *ABCG2*-related gout/hyperuricemia risk and the biochemical characteristics of the ABCG2 protein.

## 1. Introduction

Gout—a common disease with increasing global occurrence, typically causing severe pain and physical disability—results from hyperuricemia characterized by elevated serum uric acid (SUA) concentrations [1]. Indeed, uric acid accumulation in the body causes hyperuricemia and subsequently increases the risk of gout. Although not all individuals with hyperuricemia develop symptomatic gout, the risk of gout increases in proportion to the elevation of urate in circulation. Humans do not have a functional uricase (urate-degrading enzyme) [2], so uric acid is the final metabolite in the purine catabolic pathway in humans. Hence, urate excretion from the body is necessary for the maintenance of uric acid homeostasis. Moreover, accumulating evidence suggests that the net amount of excreted uric acid is regulated mainly by urate transporters, such as urate transporter 1 (URAT1, a renal urate re-absorber) [3], solute carrier family 2 member 9 (SLC2A9, also known as glucose transporter member 9) [4,5,6], and ATP-binding cassette subfamily G member 2 (ABCG2, a high capacity urate exporter expressed in the kidney and intestine) [7,8,9]. Importantly, we recently reported that decreased extra-renal urate excretion caused by ABCG2 dysfunction is a common mechanism of hyperuricemia [7,10,11].

As a 655-amino acid *N*-linked glycoprotein expressed on the renal and intestinal brush border membranes, ABCG2 transports its substrates, such as urate, from the cytosol to the extracellular space in an ATP-dependent manner [12]. The human *ABCG2* gene consists of 16 exons and 15 introns and is located on chromosome 4q21–q22 [13], a region that showed one of the most significant associations with gout susceptibility in a series of Genome-wide association studies (GWASs) [8,14,15,16,17,18,19]. Hitherto, 45 allelic variants have been found in the *ABCG2* gene with most of the variants having been reported to have wide ethnic differences in allele frequency. In contrast, two single nucleotide polymorphisms have been reported in *ABCG2*: c.34G>A (V12M) and c.421C>A (Q141K) are recognized as common variants in a relatively large number of ethnic groups [20]. Biochemical characterizations revealed that the V12M variant has no effects on the expression and urate transport activity of ABCG2, whereas the Q141K variant decreases the cellular function of ABCG2 through reduction of its protein levels [12]. In addition to Q141K, ABCG2 Q126X (c.376C>T), which is common in the Japanese population but rare in other populations, has additionally been identified as a hyperuricemia- and gout-risk allele [9,10,11,21]. Given that these two variants are associated with a significantly increased risk of gout (odds ratio > 3) [9,21], the effects of common variants of *ABCG2* on gout susceptibility are likely to be genetically strong. However, although some information is available [22,23,24], the effects of rare variants are not fully understood.

In our previous study employing genetic analyses for gout patients in Japan and functional analyses of some non-synonymous rare variants of *ABCG2*, we found that multiple rare and common variants of *ABCG2* are independently associated with gout risk [22]. Nevertheless, this “Common Disease, Multiple Common and Rare variant” model for the association between *ABCG2* and gout needs to be further validated, especially in other populations. To this point, we recently identified ten non-synonymous variants of *ABCG*2, including two common variants and eight rare variants, using a cohort of 145 patients with gout in the Czech Republic [23,24]. Among the rare variants, only an intron splicing variant c.689+1G>A that causes a frameshift-dependent premature stop codon was identified as an *ABCG2* null variant via functional assays [24]. However, regarding the other rare variants, whether each one is associated with gout risk in the Czech population remains to be elucidated. Moreover, owing to lack of molecular analyses, the previous study could not determine the effects of each rare variant on the urate transport activity of ABCG2 [23]. Importantly, except for a rare variant K360del (c.1079_1081delAGA), the rare variants identified in the Czech population [23] were not found in the Japanese population [22]. Therefore, further studies on clinical risk prediction for gout in terms of rare *ABCG2* variants as well as the functional validation of such variants are required.

In the present study, we employed an enlarged cohort of 250 patients with hyperuricemia or gout to determine non-synonymous allelic variants of *ABCG2* related to the risk of such diseases. Based on the results of the sequence analysis of *ABCG2* and database information, nine rare exonic variants of ABCG2 (R147W, T153M, K360del, F373C, T421A, T434M, S476P, S572R, D620N) were subjected to functional analyses. Here, we demonstrate the novel effects of these rare exonic variants on the expression, cellular localization, and function of ABCG2 protein as a urate transporter via molecular analyses. Our findings might support a “Common Disease, Multiple Common and Rare variant” hypothesis for the association between *ABCG2* and gout susceptibility in a European population. Additionally, these findings about population-specific genetic variants could deepen our understanding of the heritability of SUA levels and gout, which were previously estimated to be 27–41% and approximately 30%, respectively, in Europeans [14].

## 2. Materials and Methods

### 2.1. Clinical Subjects

The analyzed set consists of two groups: a hyperuricemic group consisting of 68 subjects and a gout group consisting of 182 subjects, which was an enlarged cohort compared to that containing 145 gout subjects described previously [23]. The main cohort of 58 primary hyperuricemia subjects and 177 subjects with gout was selected from patients of the Institute of Rheumatology, Prague, the Czech Republic. The 10 pediatric subjects with hyperuricemia and five with gout was selected from patients of the Department of Pediatrics and Adolescent Medicine, Charles University, Prague as previously described [25].

In terms of SUA, the definition of hyperuricemia was as follows: (1) men > 420 μmol/L on two repeated measurements taken at least 4 weeks apart and (2) women and children under 15 years > 360 μmol/L on two repeated measurements taken at least 4 weeks apart. Gouty arthritis was diagnosed according to the American College of Rheumatology criteria: (1) the presence of sodium urate crystals seen in the synovial fluid using a polarized microscope or (2) at least six of 12 clinical criteria being met [26]. Patients suffering from secondary gout and other purine metabolic disorders associated with pathological concentrations of SUA were excluded. The control group consisted of 132 normouricemic subjects, which was an enlarged control population compared to that containing 115 subjects described previously [23], was selected from among the personnel of the Institute of Rheumatology. All tests were performed in accordance with standards set by the institutional ethics committees, which approved the project in Prague (no.6181/2015).

### 2.2. PCR Amplification of ABCG2 and Sequence Analysis

*ABCG2* coding regions were analyzed from genomic DNA, as described previously [23]. The reference sequence was defined as version ENST00000237612.7 (location: Chromosome 4: 88,090,269−88,158,912 reverse strand) (www.ensembl.org). The reference protein sequence was defined as Q9UNQ0 (http://www.uniprot.org/uniprot).

### 2.3. Materials

ATP, AMP, creatine phosphate disodium salt tetrahydrate, and creatine phosphokinase type I from rabbit muscle were purchased from Sigma-Aldrich (St. Louis, MO, USA) and [8-^14^C]-uric acid (53 mCi/mmol) were purchased from American Radiolabeled Chemicals (St. Louis, MO, USA). All other chemicals used were commercially available and of analytical grade.

### 2.4. Preparation of ABCG2 Mutants’ Expression Vectors

To express human ABCG2 (NM_004827.3) fused with the EGFP-tag at its N-terminus (EGFP-ABCG2) and EGFP (control), we used an ABCG2/pEGFP-C1 plasmid that was generated in our previous study [27]. Using a site-directed mutagenesis technique, vectors expressing different ABCG2 variants were generated from an ABCG2 wild-type (WT)/pEGFP-C1 plasmid. To confirm the introduction of mutations, each variant cDNA of ABCG2 fused with EGFP generated in the plasmid was subjected to full sequencing using the BigDye Terminator v3.1 (Applied Biosystems Inc., Foster City, CA, USA) and an Applied Biosystems 3130 Genetic Analyzer (Applied Biosystems Inc.) according to the methods described in our previous study [27].

### 2.5. Cell Culture

Human embryonic kidney 293 cell-derived 293A cells were purchased from Life Technologies (Carlsbad, CA, USA) and cultured in Dulbecco’s Modified Eagle’s Medium (DMEM; Nacalai Tesque, Kyoto, Japan) supplemented with 10% fetal bovine serum (Biowest, Nuaillé, France), 1% penicillin/streptomycin, 2 mM L-glutamine (Nacalai Tesque), and 1 × Non-Essential Amino Acid (Life Technologies) at 37 °C in an atmosphere of 5% CO_2_ as described previously [27]. All experiments were carried out with 293A cells at passages 10–16.

Each vector plasmid for ABCG2 WT or its mutants was transfected into 293A cells by using polyethyleneimine MAX (PEI-MAX; 1 mg/mL in milliQ water, pH 7.0; Polysciences Inc., Warrington, PA, USA) as described previously [27] with some modifications. The amount of plasmid DNA used for transfection was adjusted to be the same for ABCG2 WT and its mutants. In brief, each plasmid was mixed with PEI-MAX (1 μg of plasmid/5 μL of PEI-MAX for 5 × 10^5^ 293A cells) in Opti-MEM^TM^ (Thermo Fisher Scientific K.K., Kanagawa, Japan) and incubated for 20 min at room temperature. 293A cells were collected after treatment with a 2.5 g/L-Trypsin and 1 mmol/L-EDTA solution (Nacalai Tesque), followed by centrifugation at 1000× *g* for 5 min. The cell pellet was re-suspended in fresh DMEM, and the resulting suspension was mixed with plasmid/PEI-MAX mixture (50:50, v/v). Then, the cells were re-seeded at a concentration of 1.4 × 10^5^ cells/cm^2^ onto a collagen-coated glass bottom dish (cover size 22 × 22 mm and 0.16–0.19 mm thick; Matsunami Glass Inc., Tokyo, Japan) for confocal microscopy or cell culture plates for whole cell lysate preparation. The medium was replaced with a fresh medium after the first 24 h of incubation.

### 2.6. Preparation of Whole Cell Lysates

At indicated periods after the plasmid transfection, whole cell lysates were prepared in an ice-cold lysis buffer A containing 50 mM Tris/HCl (pH 7.4), 1 mM dithiothreitol, 1% (w/v) Triton X-100, and a protease inhibitor cocktail for general use (Nacalai Tesque) as described previously [28]. Protein concentration of the whole cell lysate was quantified using a BCA Protein Assay Kit (Pierce, Rockford, IL, USA) with bovine serum albumin (BSA) as a standard according to the manufacturer’s protocol. For glycosidase treatment, the whole cell lysate samples were incubated with PNGase F (New England Biolabs Japan Inc., Tokyo, Japan) (1.25 U/μg of protein) at 37 °C for 10 min as described previously [29,30], and then subjected to immunoblotting.

### 2.7. Preparation of ABCG2-Expressing Plasma Membrane Vesicles

The membrane vesicles were prepared from ABCG2-expressing 293A cells as described previously [24] with minor modifications. In brief, 293A cells seeded on 145-mm tissue culture dishes (CELLSTAR, 145 × 20 mm; Greiner Japan, Tokyo, Japan; 10 dishes/variant) at approximately 80% confluency were transiently transfected with each plasmid using PEI-MAX (24 μg of plasmid/120 μL of PEI-MAX/145-mm dish). Forty-eight hours after the transfection, the cells were collected and subjected to plasma membrane isolation. The 293A cells were suspended in a hypotonic buffer (1 mM Tris/HCl, 0.1 mM EDTA, pH 7.4, and the protease inhibitor cocktail for general use) and gently stirred for 1 h on ice. Then, the solution was ultra-centrifuged at 100,000× *g* for 30 min at 4 °C, and the pellet was diluted with an ice-cold isotonic buffer (10 mM Tris/HCl, 250 mM sucrose, pH 7.4), then homogenized with a dounce tissue homogenizer (Cat#: 2-4527-03; ASONE, Osaka, Japan) on ice. The crude membrane fraction was carefully layered over a 38% sucrose solution (5 mM Tris/HEPES, pH 7.4). After ultra-centrifugation at 280,000× *g* for 45 min at 4 °C, the turbid layer at the interface was collected, suspended in the isotonic buffer, and ultra-centrifuged at 100,000× *g* for 30 min at 4 °C. The membrane fraction was collected and re-suspended in an isotonic buffer and then passed through a 25-gauge needle. The resulting plasma membrane vesicles were rapidly frozen in liquid N_2_ and kept at −80 °C until used. The protein concentration was measured using the BCA Protein Assay Kit.

### 2.8. Immunoblotting

Expression of ABCG2 protein in whole cell lysate and plasma membrane vesicles was examined by immunoblotting according to the methods reported in our previous study [31]. In brief, the prepared samples were mixed with a sodium dodecyl sulfate polyacrylamide gel electrophoresis (SDS-PAGE) sample buffer solution containing 10% 2-mercaptoethanol, separated by electrophoresis on poly-acrylamide gels, and then transferred to Polyvinylidene Difluoride membranes (Immobilon; Millipore Corporation, Billerica, MA, USA) by electroblotting at 15 V for 60 min.

For blocking, the membrane was incubated in Tris-buffered saline containing 0.05% Tween 20 and 3% BSA (Nacalai Tesque) (TBST-3% BSA) for 1 h at room temperature. Blots were probed with a rabbit anti-EGFP polyclonal antibody (A11122; Life Technologies; diluted 1000 fold in TBST 0.1% BSA), a rabbit anti-α-tubulin antibody (ab15246; Abcam Inc., Cambridge, MA, USA; diluted 1000 fold), or a rabbit anti-Na^+^/K^+^-ATPase α antibody (sc-28800; Santa Cruz Biotechnology Inc., Santa Cruz, CA, USA; diluted 1000 fold) followed by incubation with a donkey anti-rabbit immunoglobulin G (IgG)-horseradish peroxidase (HRP)-conjugated antibody (NA934V; diluted 3000 fold). HRP-dependent luminescence was developed using the ECL^TM^ Prime Western Blotting Detection Reagent (GE Healthcare UK Ltd., Buckinghamshire, UK) and detected using a multi-imaging Analyzer Fusion Solo 4^TM^ system (Vilber Lourmat, Eberhardzell, Germany). The band density was quantified using the Fusion software (Vilber Lourmat) to assess protein expression levels.

### 2.9. Confocal Laser Scanning Microscopic Observation

For confocal laser scanning microscopy, 48 h after the transfection, 293A cells were fixed with ice-cold methanol for 10 min and then washed three times with PBS (-). Then, the cells were incubated with TO-PRO-3 Iodide (Molecular Probes, Eugene, OR, USA) diluted 250 fold in PBS (-) for 10 min at room temperature. After the visualization of nuclei, the cells were washed with PBS (-) twice and then mounted in VECTASHIELD Mounting Medium (Vector Laboratories, Burlingame, CA, USA). To analyze the localization of EGFP-fused ABCG2 protein, fluorescence was detected using a FV10i Confocal Laser Scanning Microscope (Olympus, Tokyo, Japan).

To visualize plasma membranes, we used a fluorescent wheat germ agglutinin (WGA) conjugate (WGA, Alexa Fluor^®^ 594 conjugate; Thermo Fisher Scientific K.K.) according to the manufacturer’s protocol, with minor modifications. Specifically, the cells were fixed with 4% paraformaldehyde for 15 min at room temperature and then washed three times with PBS (-). Then, the cells were treated with WGA (10 μg/mL) in PBS (-) for 10 min at room temperature. After washing with PBS (-), the cells were treated with PBS (-) containing 0.02% (w/v) Triton X-100 and then subjected to TO-PRO-3 Iodide staining as described above.

### 2.10. Urate Transport Assay

The urate transport assay with ABCG2-expressing plasma membrane vesicles was conducted using a rapid filtration technique [31,32,33]. As described in our previous report [31], we used 20 μM of radiolabeled urate in reaction mixtures (10 mM Tris/HCl, 250 mM sucrose, 10 mM MgCl_2_, 10 mM creatine phosphate, 1 mg/mL creatine phosphokinase, 0.25 mg/mL each plasma membrane vesicle, pH 7.4, and 50 mM ATP or AMP as the absence of ATP) which were incubated for 10 min at 37°C for the evaluation of ABCG2 function as an ATP-dependent urate transporter. The urate transport activity was calculated as an incorporated clearance defined as the incorporated level of urate [DPM/mg protein/min]/urate level in the incubation mixture [DPM/μL]. ATP-dependent urate transport was calculated by subtracting the urate transport activity in the absence of ATP from that in the presence of ATP. Furthermore, ATP-dependent urate transport activities of each ABCG2 variant were described as the percent of the activity of ABCG2 WT.

In addition, because urate uptake assays for the identified ABCG2 variants were conducted using a *Xenopus leavis* oocyte expression system, the relevant information is shown in the Appendix A [34].

### 2.11. Schematic Illustration of ABCG2 Protein

According to the structure of human ABCG2, which was determined by cryo-electron microscopy [35], 2D topology of ABCG2 protein was constructed, and the obtained-topology data were plotted and modified using the T(E)Xtopo package [36]. Information of highly conserved peptide motifs among ABC transporters, such as Walker A, Walker B, and signature C, in ABCG2 protein were obtained from a previous report [37].

### 2.12. Statistical Analysis

Clinical data were summarized as absolute and relative frequencies for categorical variables and as medians with interquartile range (IQR) and data range for continuous variables. Hyperuricemic and gout sub-cohorts were compared using either Fisher’s exact test for categorical variables or Wilcoxon sum-rank test for continuous variables. Kruskall-Wallis test (one-way non-parametric ANOVA) and pairwise Wilcoxon tests with Bonferroni correction (for post-hoc comparisons) were used to compare ages of gout/hyperuricemia onset among groups with zero, one, and two allelic variants of interest. Fisher’s exact test was used to compare the family gout history and the presence of alleles of interest. For individual allelic variants, minor allele frequency (MAF) data were excreted from the public databases NCBI and ExomAc and compared to cohort MAF in this study using a binomial test. Statistical language and environment R (version 3.5.0) (R Foundation for Statistical Computing, Vienna, Austria) was used for clinical data analyses.

Regarding experimental results, all statistical analyses were performed by using EXCEL 2013 (Microsoft Corp., Redmond, WA, USA) with Statcel3 add-in software (OMS publishing Inc., Saitama, Japan) according to our previous study [38]. The specific statistical tests that were used for individual experiments are described in the figure legends. Statistical significance was defined in terms of *P* values less than 0.05 or 0.01.

## 3. Results and Discussion

### 3.1. Subjects

The main demographic and biochemical characteristics of the subjects are summarized in Table 1. Our cohort (a total of 250 patients recruited from the Czech Republic) consisted of 182 individuals with primary gout (166 male/16 female) among which 66 patients (36% of the cohort) had a positive family history of gout. In a sub-cohort of 68 hyperuricemia patients (48 male/20 female), 31 patients (46% of the sub-cohort) had a positive family history. With respect to medications, 58 patients (23%) did not take any urate-lowering therapy medication, 175 patients (70%) took allopurinol, and 17 patients (7%) took febuxostat (the details are summarized in Appendix A). Considering that *ABCG2* is one of the most influential genetic risk factors for gout and hyperuricemia, we next examined a relationship between non-synonymous mutations in *ABCG2* and the risk of such diseases in our gout/hyperuricemia cohort.

### 3.2. Identification of ABCG2 Variants in a Gout/Hyperuricemia Cohort

To determine non-synonymous allelic variants in *ABCG2* relating to the risk of gout or hyperuricemia, we performed targeted exon sequencing of *ABCG2* in our cohort. The results—all identified variants and their allele frequencies—are summarized in Table 2. We identified 11 exonic non-synonymous variants including two common variants: p.V12M (rs2231137) and p.Q141K (rs2231142), which are well-characterized, nine rare variants: p.R147W (rs372192400), p.T153M (rs753759474), p.F373C (rs752626614), p.T421A (rs199854112), p.T434M (rs769734146), p.S476P (not annotated), p.S572R (rs200894058), p.D620N (rs34783571), and a three-base deletion p.K360del (rs750972998). The *ABCG2* genotype frequency of our gout/hyperuricemia cohort was compared to MAF data from the Exome Aggregation Consortium (http://exac.broadinstitute.org/) and 1000 Genomes (http://www.internationalgenome.org/), which addressed a European-origin population (Table 2). The frequency of Q141K (a common variant linked to the risk of gout/hyperuricemia [9,10,11,21]) in our cohort (0.238)—0.247 in the gout sub-cohort (87 heterozygotes and 16 homozygotes) and 0.213 in the hyperuricemia sub-cohort (19 heterozygotes and 5 homozygotes)—was significantly higher than that was reported in the European-origin population (0.094). Several rare variants were present at higher allele frequencies in our cohort compared with the control population as shown in Table 2. However, as a limitation in this study, we note that our cohort population was not sufficiently large for a detailed analysis of the individual effect of each rare variant, owing to very low MAF on the disease risk. To overcome this limitation, analysis of a larger data set will be needed in the future. 

Next, we focused on the combination of the common and rare variants of *ABCG2*. In total, 115 gout/hyperuricemia patients (46.0% of case) harbored at least one of the identified 11 non-synonymous variants (Appendix A). Among the patients, 23 individuals harbored two non-synonymous variants: 16 patients were homozygous for Q141K while seven patients had Q141K and one of the other variants. No patients harbored three or more of the non-synonymous variants, and no participants were found to be homozygous for rare variants. Interestingly, we found an association between the number of non-synonymous variants in *ABCG2* and the age of onset of gout/hyperuricemia in our cohort (Appendix A). The median age of onset among patients with zero, one, or two variants were 42, 40, and 22 years, respectively (*P* < 0.0002, Kruskal-Wallis test). Post-hoc analysis revealed that the group with two variants significantly differs from the other groups, suggesting that an increased number of non-synonymous alleles might cause an earlier age of onset.

Regarding family history, patients with non-synonymous variants had familial gout in 54 of 111 cases (48.6%), whereas patients who did not have non-synonymous variants had familial gout in 43 of 130 (33%) cases (Figure 1). This association was statistically significant (odds ratio = 1.91, 95% CI: 1.10, 3.34; *P* = 0.0176, Fisher’s exact test). Considering that a previous study with a small cohort had found only borderline significance (*P* = 0.053) for this association [23], our result can be considered to be a strength of this study. As previously described, these findings suggest the epidemiological importance of *ABCG2* common and rare variants for gout/hyperuricemia risk in the Czech population, supporting a “Common Disease, Multiple Common and Rare variant” hypothesis for the association between *ABCG2* and gout, which was proposed in our previous study on Japanese gout patients [22].

Importantly, regarding the identified rare variants of ABCG2, T421A was a novel variant and the effects of the other variants on the protein function of ABCG2 as a urate transporter have not yet been characterized. Thus, we conducted a series of functional analyses to assess the rare variants that we identified (Table 2); the positions of each mutant are described in Figure 2.

### 3.3. Effect of Each Mutation on the Glycosylation Status of ABCG2 Protein

To investigate the effect of each non-synonymous mutation in the *ABCG2* gene on the intracellular processing and function of ABCG2 protein, we transiently expressed ABCG2 WT and its nine variants (R147W, T153M, K360del, F373C, T421A, T434M, S476P, S572R, and D620N) in 293A cells. Each expression vector was prepared using a site-directed mutagenesis technique from an ABCG2 WT/pEGFP-C1 plasmid and confirmed by DNA sequence. To address the former topic, we first performed immunoblot analyses using the anti-EGFP antibody for the detection of EGFP-tagged ABCG2. The results revealed that the R147W and S572R variants did not produce a matured glycoprotein (Figure 3A). Moreover, using *N*-glycosydase (PNGase F)-treated whole cell lysates, we compared the levels of ABCG2 protein in the cells among the variants and WT (Figure 3B). Since the Q141K variant reportedly reduces ABCG2 protein level [39,40,41], we employed this variant as a control in this study. The reducing effect of Q141K on the ABCG2 protein level detected in the present study (Figure 3) was almost comparable to that in a previous study [41] which, like us, also used a similar transient expression strategy to validate *ABCG2* mutations in vitro. This consistency supports the reliability of the following results. Semi-quantitative evaluation of immunoblot signals showed that each variant decreased the levels of ABCG2 protein; the R147W and S572R variants showed significant effects (<25% of WT), the T153M and F373C variants had profound effects on reducing ABCG2 protein level (<50% of WT), and the other variants only mildly affected the protein expression of ABCG2.

### 3.4. Effect of Each Mutation on the Intracellular Localization of ABCG2 Protein

We next investigated the effect of each mutation on the intracellular localization of the ABCG2 protein. Confocal microscopic observation demonstrated that ABCG2 WT and most of the variants localized on the plasma membrane of the cells (Figure 4A). In contrast, the R147W and S572R variants that hardly experienced protein maturation processing in the cells were not localized on the plasma membrane. This result was supported by high magnification image observation under a fluorescent WGA (a plasma membrane probe)-treated condition (Figure 4B). Thus, we concluded that the R147W and S572R variants have little function as a urate transporter on the plasma membrane and we performed further analyses for the other seven variants, as described below.

### 3.5. Effect of Each Mutation on the Urate Transport Activity of ABCG2 Protein

To investigate the seven ABCG2 variants that localized on the plasma membrane, we performed functional assays using ABCG2-expressing plasma membrane vesicles. Prior to the assay, the expression of each ABCG2 variant on the plasma membrane vesicles was confirmed by immunoblot analysis (Figure 5A), which supported the results obtained through confocal microscopy (Figure 4A). Then, the ABCG2 function was evaluated as an ATP-dependent urate transport into the vesicles (Figure 5B). The results show that the T434M and S476P variants diminished the function of the ABCG2 protein. Although the T153M and F373C variants-expressing plasma membrane vesicles exhibited reduced activity of urate transport as compared with ABCG2 WT (Figure 5B), this difference in the ABCG2 function between WT and these two variants was rescued by the normalization of urate transport activity by ABCG2 protein expression on plasma membrane vesicles (Appendix A). These results suggest that the T153M and F373C variants do not affect ABCG2 function qualitatively (via alteration of its intrinsic transporter activity), but rather do so quantitatively (via decreasing its cellular protein level). The other ABCG2 variants (K360del, T421A, and D620N) had little effect on ABCG2 function. Additionally, supportive results were obtained from the *Xenopus oocyte* expression system.

### 3.6. Integration of the Obtained Data

Finally, we integrated the results of functional analyses (Figure 3, Figure 4 and Figure 5) and classified the nine ABCG2 variants according to their effects on the intracellular processing and protein function of ABCG2 (Appendix A). As summarized in Table 3, five of the eight rare variants of ABCG2 found in our gout/hyperuricemia cohort were less functional or null. Although the available information was limited, the allele frequencies of such less functional or null variants in the cohort tended to be higher than those in the European-origin population (Table 2). To this point, there was little inconsistency between the results of functional analyses and genetic analyses. However, a more comprehensive understanding requires further studies with a larger data set.

Regarding three rare variants—K360del, T421A, and D620N—that were functionally comparable to WT, their allele frequencies in our cohort were not significantly different from those in the European-origin population (Table 2). Thus, at least in their individual cases, these rare variants might have smaller effects on gout/hyperuricemia risk compared with the other identified rare variants. To further elucidate whether the presence of such functional rare variants could affect disease risk, future studies will be required to assess enlarged cohorts in terms of haplotype. With respect to biochemistry, our results indicated that K360, T421, and D620 are not essential for the function of the ABCG2 protein. With respect to D620N, a previous study, using an insect cell expression system, reported that this amino acid substitution had little effect on the ABCG2-mediated transport of porphyrin, which is an endogenous substrate of ABCG2 [37]; a similar result was obtained in this study, supporting the adequacy of functional validation we carried out. Considering that these three amino acid positions (K360, T421, and D620) are not predicted to be located in the transmembrane domains of ABCG2 (Figure 2), these positions could have a flexibility in the kind of amino acids and little effect on the appropriate structures of the ABCG2 transporter during its molecular function on the plasma membrane.

Additionally, our findings provide insights into the amino acid positions that are important for ABCG2 function. Interestingly, in null or approximately null variants of ABCG2, the original amino acids (R147, T434, S476, and S572) are conserved with several major mammalian species (Figure 6). Among them, R147 is close to Q141, while the others are in transmembrane domains (Figure 2). Given that the Q141K and R147W variants decreased the protein levels of ABCG2 (Figure 3), peptides around these two amino acids might be important for the stabilization of ABCG2 protein. Since the S572R variant likewise diminished ABCG2 protein levels (Figure 3), the S572 residue, which is located in the boundary region between helices 5b and 5c (Figure 2), may be important. Interestingly, a previous study reported that an artificial mutation I573A (a neighbor position of S572) disrupted maturation and reduced plasma membrane localization of ABCG2 protein [42], which could support the biochemical importance of the boundary region in the intracellular stability of the ABCG2 protein. Regarding T434 and S476, there is a possibility that these amino acids might be involved in the formation of a penetrating pathway for ABCG2 substrates or may be critical for the dynamics of the function of the ABCG2 transporter.

Before closing, we would like to discuss the necessity for the analyses of rare *ABCG2* variants. Based on the recent findings of GWASs of clinically defined gout [16,43], common variants of *ABCG2* are extremely important in gout pathogenesis. However, because the *ABCG2* gene is reportedly to be highly-polymorphic with population specificity [44], there will be population-specific rare variants of *ABCG2* that could be a genetic risk factor of gout. Indeed, such rare variants that were found in the present study with the Czech Republic population were distinct from those found in our previous study with a Japanese population [22]. Considering the theoretical limitations in GWAS analyses, which focus on only single nucleotide polymorphisms (SNPs), the clinical and experimental approaches that we employed in this study could be a reasonable way to identify pathophysiologically important rare variants. Given that many SNPs in *ABCG2* have been studied [20], identification and validation of population-specific rare variants will be important for achieving more effective and accurate prediction of ABCG2-related gout/hyperuricemia risk.

## 4. Conclusions

In the present study, we explored the exonic non-synonymous variants of *ABCG2* using a cohort of 250 individuals (68 primary hyperuricemia patients and 182 primary gout patients) recruited from a European descent population in the Czech Republic. Patients with non-synonymous variants showed an earlier on set of gout (Appendix A), which is consistent with the results of previous studies [21,23]. The enlarged cohort enabled us to reveal that the numbers of non-synonymous variants of *ABCG2* could affect the frequency of familial gout (Figure 1), which was inconclusive in our previous study because of the small sample size [23]. Moreover, as summarized in Table 3, we successfully characterized the nine rare variants of ABCG2 (Figure 3, Figure 4 and Figure 5). Additionally, given that ABCG2 is recognized to be an important determinant of the pharmacokinetic characteristics of its substrate drugs [45,46], this information will be significant for the field of pharmacogenomics.

In summary, our findings will deepen our understanding of *ABCG2*-related gout/hyperuricemia risk as well as the biochemical characteristics of the ABCG2 protein. To achieve a more accurate evaluation of an individual’s risk for gout, addressing rare *ABCG2* variants is of importance. Furthermore, for effective genotyping in clinical situations, uncovering the population-specificities of such rare variants will be important.

## Figures and Tables

**Figure 1 cells-08-00363-f001:**
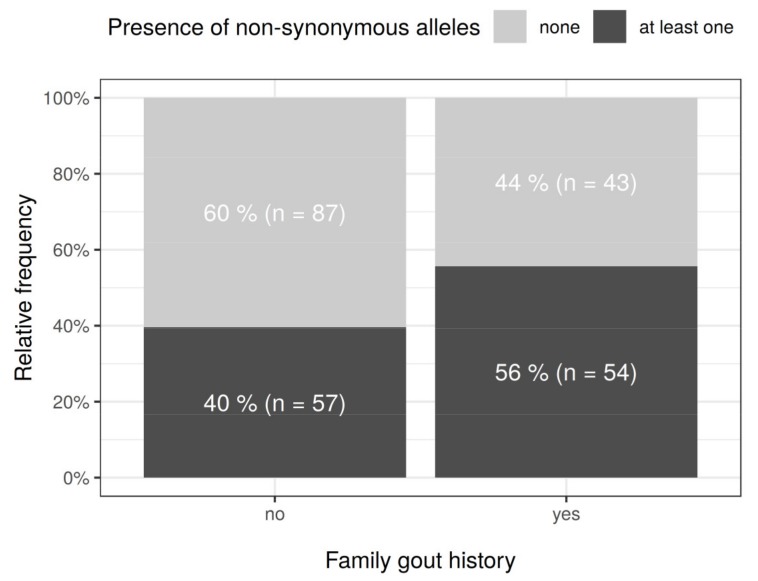
Family history of gout and the numbers of allelic variants in *ABCG2*. Depending on the presence or absence of family gout history, proportion of gout patients with or without any of the 11 non-synonymous alleles identified in *ABCG2* is summarized. The presence of *ABCG2* allelic variants was associated with the gout family history (odds ratio = 1.91, 95% CI: 1.10, 3.34; *P* = 0.0176, Fisher’s exact test).

**Figure 2 cells-08-00363-f002:**
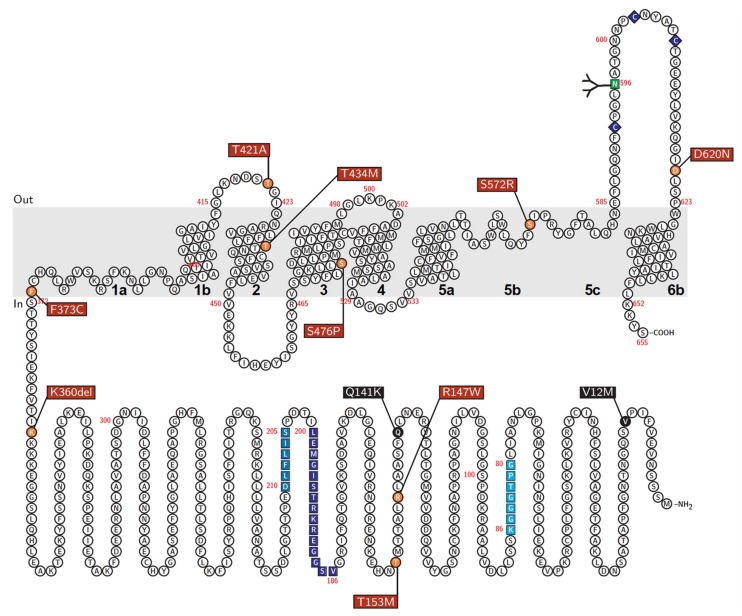
Schematic illustration of a putative topological model of human ABCG2 protein. Red box, rare variants analyzed in the present study; *Black box*, common variants. Helices in the transmembrane domain are numbered (1a to 6b) according to a previous study [34]. Asn596 is an *N*-linked glycosylation site. Unique motifs common to ABC proteins: Walker A (amino acids 80–86), Walker B (amino acids 205–210), and signature C (amino acids 186–200) in ABCG2 protein are indicated by colors.

**Figure 3 cells-08-00363-f003:**
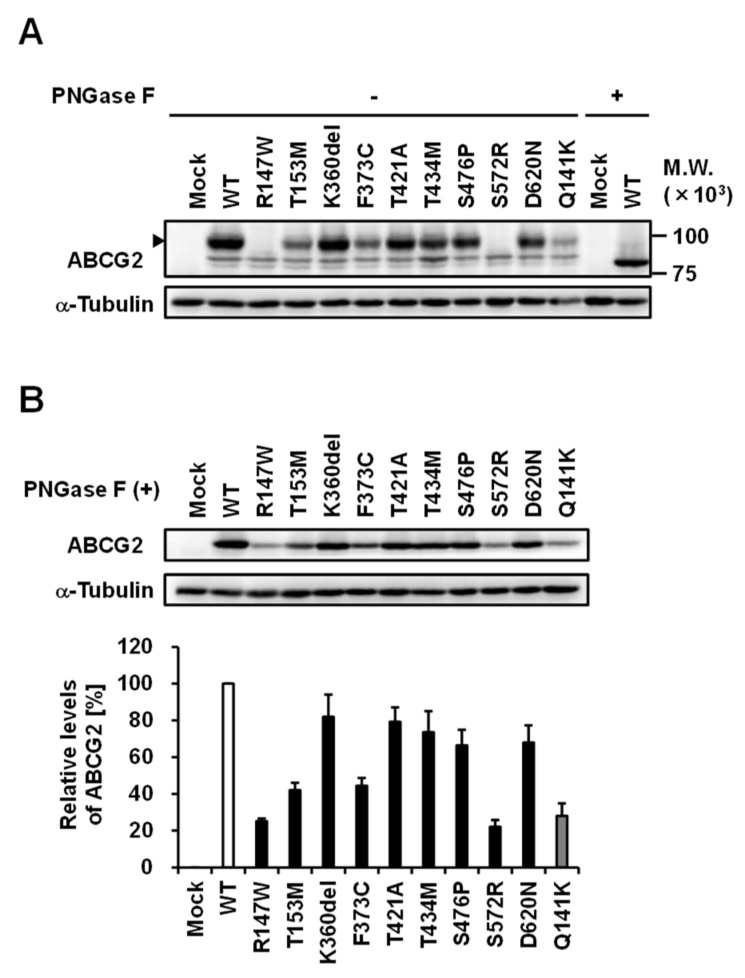
Effects of each mutation on the maturation status and protein levels of ABCG2 in transiently transfected 293A cells. (**A**) Immunoblot detection of ABCG2 wild-type (WT) and its variants in the whole cell lysate samples that were prepared 48 h after the transfection. Arrowhead, matured ABCG2 as a glycoprotein; α-Tubulin, a loading control. (**B**) Relative protein levels of ABCG2 WT and its variants. The signal intensity ratio (ABCG2/α-tubulin) of the immunoreactive bands was determined and normalized to that in ABCG2 WT-expressing cells. Data are expressed as the mean ± SD. *n* = 3.

**Figure 4 cells-08-00363-f004:**
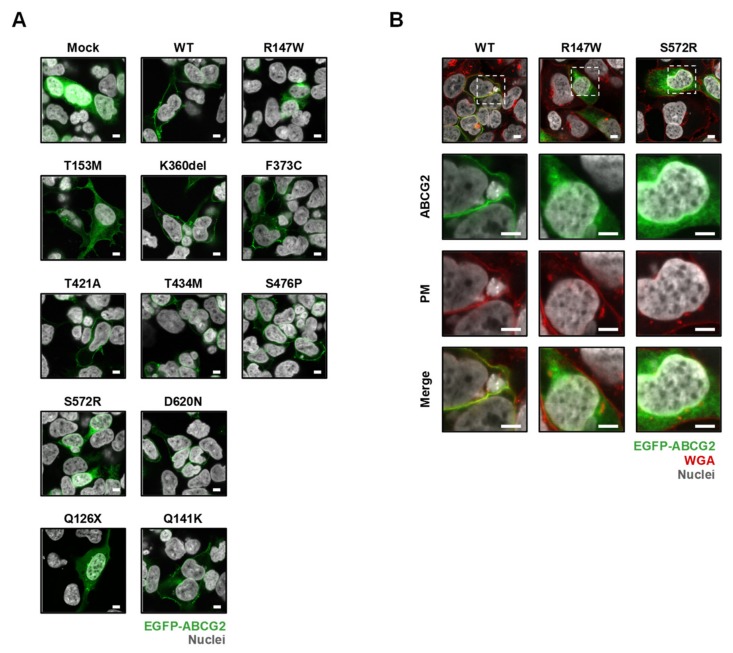
Effects of each mutation on the cellular localization of ABCG2 protein in transiently transfected 293A cells. (**A**) Intracellular localization of ABCG2 variants. Q126X (a stop gain variant that is deficient in the plasma membrane localization [22]) and Q141K are controls. (**B**) High magnification images of cells transfected with R147W and S572R variants indicate that these mutations impaired localization to the plasma membrane of the cells. Framed areas in the panels of top lane were observed under a higher magnification. Confocal microscopic images were obtained 48 h after the transfection. Nuclei were stained with TO-PRO-3 iodide (gray). Plasma membrane (PM) was labeled with Alexa Fluor^®^ 594-conjugated wheat germ agglutinin (WGA) (red). Bars indicate 5 μm.

**Figure 5 cells-08-00363-f005:**
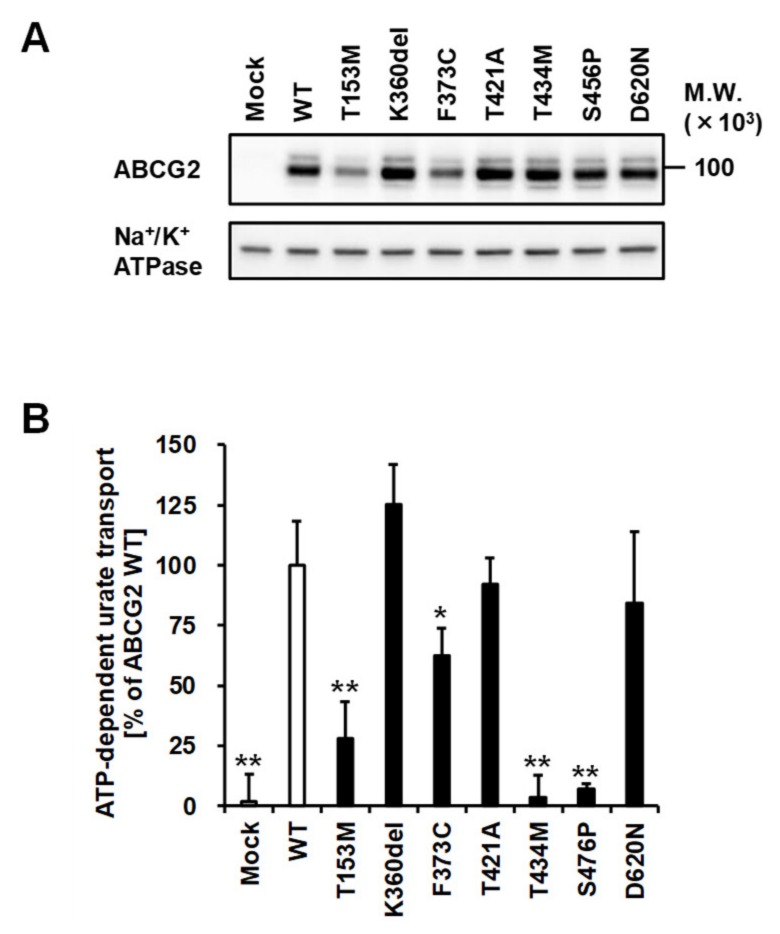
Functional validation of each ABCG2 variant as ATP-dependent urate transporters. (**A**) Immunoblot detection of ABCG2 WT and its variants expressed in plasma membrane vesicles prepared from 293A cells. Na^+^/K^+^ ATPase, a loading control. (**B**) ATP-dependent transport of urate by ABCG2 WT and its variants. The data are shown as % of WT; data are expressed as the mean ± SD. *n* = 3. Statistical analyses for significant differences were performed using Bartlett’s test, followed by a Dunnett’s test (*, *P* < 0.05; **, *P* < 0.01 *vs* WT).

**Figure 6 cells-08-00363-f006:**
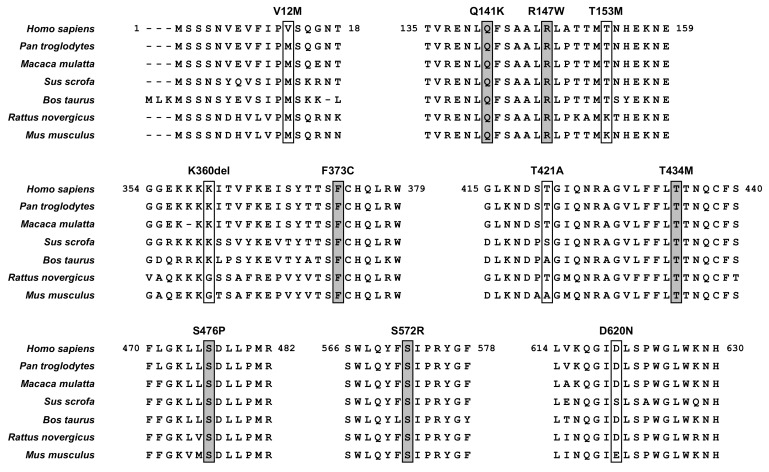
ABCG2 amino acids evolutionary conserved among seven mammalian species. The positions of non-synonymous substitutions conserved among seven species examined in the present study are grey labelled. Regarding Abcg2 protein in each species, NCBI Reference Sequence ID and amino acid sequence identity (*vs* human ABCG2, NM_004827.3) are summarized as below: *Pan troglodytes* (Chimpanzee, GABE01009237.1), 99%; *Macaca mulatta* (Rhesus macaque, NM_001032919.1, 96%; *Sus scrofa* (Pig, NM_214010.1), 84%; *Bos taurus* (Bovine, NM_001037478.3), 84%; *Rattus novergicus* (Rat, NM_181381.2), 81%; *Mus musculus* (Mouse, NM_011920.3), 81%. Multiple sequence alignments and homology calculations were carried out using the GENETYX software (GENETYX Co., Tokyo, Japan) with the ClustalW2.1 Windows program according to our previous study [27].

**Table 1 cells-08-00363-t001:** Demographic, biochemical, and genetic characteristics of gout and hyperuricemia cohorts.

**Characteristic**		**All Patients** **(*N* = 250)**			**Gout Patients** **(*N* = 182)**			**Hyperuricemia Patients** **(*N* = 68)**			***P*-Value ^#^**
		***N***	**%**		***N***	**%**		***N***	**%**		
Gender	Male	214 36	85.6 14.4		166 16	91.2 16.8		48 20	70.6 29.4		0.0002
	Female										
Familial occurrence		97	38.8 (40.2 *)		66	36.3 (36.5 *)		31	45.6 (51.7 *)		0.0480
**Characteristic**		***N***	**Median** **(IQR)**	**Range**	***N***	**Median** **(IQR)**	**Range**	***N***	**Median** **(IQR)**	**Range**	***P*-Value ^†^**
Age at examination [years]		250	51.5 (25.0)	3–90	182	54.0 (21.0)	11–90	68	36.0 (42.0)	3–78	<0.0001
BMI at examination		209	28.4 (5.8)	16–50	151	28.4 (5.4)	19.5–50	58	28.1 (6.4)	16–41	0.0822
Gout/hyperuricemia onset ^$^ [years]		236	40.0 (28.0)	1.2–84	181	40.0 (24.0)	8–84	55	27.0 (40.5)	1.2–76	0.0070
SUA at examination, with medication [µmol/L]		201	375.0 (134.0)	163–808	159	372.0 (128.0)	163–808	42	424.0 (140.0)	240–628	0.0515
FEUA at examination, with medication		194	3.4 (2.0)	0.9–14	158	3.4 (1.9)	0.9–14	36	3.8 (2.1)	1.3–8	0.5862

^#^ Fisher’s exact test for categorical and ^†^ Wilcoxon two-sample sum rank test were used to compare the gout sub-group with the hyperuricemia sub-group; * relative frequencies when missing information about familial occurrence was excluded; ^$^ onset (gout) and age of ascertainment (hyperuricemia). IQR, interquartile range; SUA, serum uric acid; FEUA, fractional excretion of uric acid.

**Table 2 cells-08-00363-t002:** Identified ABCG2 variants and their mutant allele frequency.

ABCG2 Variants (rs Number)	Gout (*N* = 182)		Hyperuricemia (*N* = 68)		All Patients (*N* = 250)			Normouricemia (*N* = 132)		Population MAF	*P*-Value ^#^
	*N*	MAF	*N*	MAF	*N*	MAF	95% CI *	*N*	MAF		
***Common***											
V12M (rs2231137)	8	0.0220	1	0.0074	9	0.0180	0.0083, 0.0339	5	0.0189	0.0610	<0.0001
Q141K (rs2231142)	90	0.2473	29	0.2132	119	0.2380	0.2013, 0.2778	22	0.0833	0.0940	<0.0001
***Rare***	***N***		***N***		***N***	**MAF**		***N***		**Population MAF**	
R147W (rs372192400)	1		0		1	0.0020		1		0.0001	
T153M (rs753759474)	1		0		1	0.0020		0		0.0001	
F373C (rs752626614)	1		0		1	0.0020		0		0.0000	
T421A (rs199854112)	0		1		1	0.0020		0		0.0001	
T434M (rs769734146)	1		1		2	0.0040		1		0.0000	
S476P (Not annotated)	1		0		1	0.0020		0		*No data*	
S572R (rs200894058)	1		0		1	0.0020		0		0.0002	
D620N (rs34783571)	2		0		2	0.0040		0		0.0040	
K360del (rs750972998)	1		0		1	0.0020		0		0.0001	

For common variants: * A 95% confidence interval (CI) for minor allele frequency (MAF) was estimated; ^#^ Binomial test was used (all patients vs population control). Comparative information on the obtained MAFs for the cases in this study versus those for the 132 normouricemic subjects in the control group from the Institute of Rheumatology (Prague, Czech Republic) is shown. For rare variants: due to very small counts of each rare variant, MAF for the whole sample of 250 patients was given as well as population MAF (if available).

**Table 3 cells-08-00363-t003:** Summary of the effects of each mutation on ABCG2 protein and its function.

rs Number	Nucleotide Change	AA Change	PM Localization	Protein Level on PM	Urate Transport	Effect on the Cellular ABCG2 Function
rs372192400	439C>T	R147W	−	N.D.	N.D.	Null
rs753759474	458C>T	T153M	+	+	+	Decrease to about a quarter
rs750972998	1079_1081delAGA	K360del	+	++	++	N.S.
rs752626614	1118T>G	F373C	+	+	++	Decrease to about half
rs199854112	1261A>G	T421A	+	++	++	N.S.
rs769734146	1301C>T	T434M	+	++	−	Almost null
Not annotated	1426T>C	S476P	+	++	−	Almost null
rs200894058	1714A>C	S572R	−	N.D.	N.D.	Null
rs34783571	1858G>A	D620N	+	++	++	N.S.

The effects in each column are relatively indicated by: ++, comparable to WT; +, positive in localization or less than ++; −, negative (disruption). AA, amino acid; PM, plasma membrane; N.D., not determined; N.S. not significantly different.

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
