# Peer review of "Functional Characterization of Clinically-Relevant Rare Variants in ABCG2 Identified in a Gout and Hyperuricemia Cohort"

_cells, 2019, doi:10.3390/cells8040363_

Round 1
Reviewer 1 Report
The authors employed a cohort of 250 Czech individuals of European descent to study the association of allelic variants of ABCG2 with the risk of hyperuricemia and gout. Nine rare variants were identified and functionally characterized in a heterologous expression system to study the impact of the variations on the expression levels, localization and transport activity of ABCG2 protein.
The manuscript is very easy to follow and is very nicely written. The experiments are well designed and the conclusions are well supported by the data presented. This is a very solid study, which makes a significant contribution in our understanding of the structure-function relationship of this important efflux transporter and its association with the risk of gout. I recommend accepting this manuscript in its present form.
Author Response
Thank you for your letter and for the positive comments.
Reviewer 2 Report
The authors present a very nice piece of collaborative work to shed more light on the functional correlation between variants of ABCG2 and urate transport/hyperuricaemia/gout.
The majority of the literature to date has focussed on the gout associated SNP Q141K and the current paper, rather than focussing on this, analyses some rare variants of ABCG2 at both the clinical level and the functional level. There are some important findings but I have several suggestions – one of which I think is important regarding some control data – before the paper should be published.
1. Minor suggestion: The clinical data is nicely presented and the caveats associated with interpretation from rare variants are made clear. However, I do think that in Table 2 the statistical information for the rare variants should simply be omitted. The authors described between lines 284 and 289 why their data should not be interpreted with statistics and so I think the table would better reflect their own conclusions if they were removed (except the for the common SNPs)
2. Minor suggestion: the localisation and expression data is very interesting, and even in transiently transfected cells it is obvious that the mutants have different effects. The S572R mutant is intriguing because an adjacent residue (I573) also gives a trafficking defect when mutated in vitro (see Haider et al., 2015). Does this suggest that this capping helix region is important in protein folding?
3. Minor suggestion: although the membrane localisation by confocal microscopy is adequate I think it would have been better in live cells, without the co-staining. The fixing process can damage the membrane and lead to artefacts. I don’t think the authors are seeing that as their data from confocal agrees nicely with their other data but if they had live cell data it might be worth mentioning this.
4. Minor: in two places (e.g. line 383 and line 227) the authors promise data from a xenopus oocyte expression system, and they even give an author credit for this work (line 483). This data is missing from the pdf I downloaded. Please can the authors either amend the paper for its absence, or ensure this data is present.
5. Major: the functional data on membrane vesicles is well described and clearly shows that, in comparison to wild type, three mutants have an altered transport. T434M and S476P look to be severely affected and T153M is moderately affected. The latter mutant also shows a lower expression so I accept the authors’ correction for expression level which indicates that T153M is functional. My concern here is that the Q141K variant was not tested in this assay. This provides the perfect control “in my book”. It has reduced expression in other studies so it should definitely be included here as a comparison for mutants which may be loss of function and/or expression. I think data for Q141K in membrane vesicles should be included.
6. Minor: I am not convinced by the need for both Table 3 and Figure 5; they say exactly the same thing. Table 3 is preferred and figure 5 could be removed.
Author Response
The authors thank the reviewers for their time and efforts of careful reading of our manuscript. We greatly appreciate the reviewers’ constructive comments. All the changes made in the revised manuscript are marked in red.
1. Minor suggestion: The clinical data is nicely presented and the caveats associated with interpretation from rare variants are made clear. However, I do think that in Table 2 the statistical information for the rare variants should simply be omitted. The authors described between lines 284 and 289 why their data should not be interpreted with statistics and so I think the table would better reflect their own conclusions if they were removed (except the for the common SNPs)
Answer:
Thank you for your comments. According to this advice, we revised the main text and Table 2 as follows.
Revised text (Line 282-287):
Several rare variants were present at higher allele frequencies in our cohort compared with the control population as shown in Table 2. However, as a limitation in this study, we note that our cohort population was not sufficiently large for a detailed analysis of the individual effect of each rare variant owing to very low MAF on the disease risk. To overcome this limitation, analysis a larger data set will be needed in the future.
Revised Table 2 and Table legend (in file)
2. Minor suggestion: the localisation and expression data is very interesting, and even in transiently transfected cells it is obvious that the mutants have different effects. The S572R mutant is intriguing because an adjacent residue (I573) also gives a trafficking defect when mutated in vitro (see Haider et al., 2015). Does this suggest that this capping helix region is important in protein folding?
Answer:
We really acknowledged this valuable input. Considering the suggested reference, we added some discussions in our revised manuscript.
Revised text (Line 429-434):
Since the S572R variant likewise diminished ABCG2 protein levels (Figure 2), the S572 residue, which is located in boundary region between helices 5b and 5c (Figure 1), may be important. Interestingly, a previous study reported that an artificial mutation I573A (a neighbor of S572) disrupted maturation and reduced plasma membrane localization of ABCG2 protein [39], which could support the biochemical importance of the boundary region in the intracellular stability of ABCG2 protein.
New reference 39
Haider, A.J.; Cox, M.H.; Jones, N.; Goode, A.J.; Bridge, K.S.; Wong, K.; Briggs, D.; Kerr, I.D. Identification of residues in ABCG2 affecting protein trafficking and drug transport, using co-evolutionary analysis of ABCG sequences. Biosci Rep 2015, 35, doi:10.1042/BSR20150150
3. Minor suggestion: although the membrane localisation by confocal microscopy is adequate I think it would have been better in live cells, without the co-staining. The fixing process can damage the membrane and lead to artefacts. I don’t think the authors are seeing that as their data from confocal agrees nicely with their other data but if they had live cell data it might be worth mentioning this.
Answer:
Thank you for your positive feedback. Considering the followings, there seems to be little possibility of artifact. First, according to a previous study (Orbán TI et al. Biochem Biophys Res Commun. 2008 Mar 14;367(3):667-73), fixing processes hardly affected the observed locarization of EGFP-ABCG2 protein in HEK293 cells. In the previous study, there was little difference in the detected signals corresponding to EGFP-ABCG2 between the immunohistochemistry (fixed) samples and live imaging samples. Second, regarding WT, R147W, and S572R, fixing processes (methanol in Fig 3A; 4% PFA in Fig 3B) hardly affected the detected signals in the present study. Thrid, we successfully detected the intracellular localization of Q126X (a stop gain variant that is deficient in the plasma membrane localization). Regarding the last point, we revised the Figure 3A; we added Q126X as follows.
Revised Figure 3 (in file) and Figure legend 3:
(A) Intracellular localization of ABCG2 variants. Q126X (a stop gain variant that is deficient in the plasma membrane localization [22]) and Q141K are controls.
Moreover, regarding Mock, WT, R147W, and S572R, we have obtained similar results in our additional experiments for live imaging using a fluorescence microscopy which were carried out in this revise period (Figure #1 for Reviewer #2).
Figure #1 for Reviewer #2 (in file)
Legend for Figure #1 for Reviewer #2: Forty-eight hours after the transfection of each ABCG2-expressing plasmid, 293A cells without any fixing processes were observed using a fluorescence microscopy
4. Minor: in two places (e.g. line 383 and line 227) the authors promise data from a xenopus oocyte expression system, and they even give an author credit for this work (line 483). This data is missing from the pdf I downloaded. Please can the authors either amend the paper for its absence, or ensure this data is present.
Answer:
The data from Xenopus oocytes was uploaded on the Journal system as an independent manuscript-supplementary.zip file. We suppose that the reviewer #2 downloaded the manuscript.pdf file, while he/she missed the zip file. Since the reviewer #1 and #2 did not pointed out this matter, we believe that the data is present on the system, so please check whether the zip file can be obtained and seen.
5. Major: the functional data on membrane vesicles is well described and clearly shows that, in comparison to wild type, three mutants have an altered transport. T434M and S476P look to be severely affected and T153M is moderately affected. The latter mutant also shows a lower expression so I accept the authors’ correction for expression level which indicates that T153M is functional. My concern here is that the Q141K variant was not tested in this assay. This provides the perfect control “in my book”. It has reduced expression in other studies so it should definitely be included here as a comparison for mutants which may be loss of function and/or expression. I think data for Q141K in membrane vesicles should be included.
Answer:
We are very happy to hear that you may follow our studies because, to the best of our knowledge, there are only two papers (Science Translational Medicine, 2009, Reference 9; RMD open, 2017, Reference 22) that directly compared the urate transport activities of ABCG2 between WT and Q141K variant using mamalian cells-derived plasma membrane vesicles (one of the standard methods of functional assay for ABC transporters) and these papers are our studies. We agree that the Q141K could be a good experimental control in functional assays; however, regarding your concern described above, we convinced that information on Q141K is not essential in the present study because there have already been both positive (WT) and negative (mock) controls in our vesicle transport assay (Figure 4). In other words, we successfully detected the urate transport activities of ABCG2. Furthermore, our result about D620N was consistent with a previous study (see Line 416-419), which can assure the adequacy of our experiments. To emphasize this point, we added a sentence in the revised manuscript.
Revised text (Line 416-419):
With respect to D620N, a previous study using an insect cell expression system reported that this amino acid substitution had little effect on the ABCG2-mediated transport of porphyrin, which is an endogenous substrate of ABCG2 [35]; the similar result was obtained in this study, supporting the adequacy of functional validation we carried out.
6. Minor: I am not convinced by the need for both Table 3 and Figure 5; they say exactly the same thing. Table 3 is preferred and figure 5 could be removed.
Answer:
Thank you for your kind suggestion. According to this comment, we removed the initial Figure 5 from main items. Since we think that this Figure will be a help to understand the summary of functional validation visually, we have used it as a Supplementary Figure A3. Additionally, the initial Figure 6 was re-named as Figure 5 in the revised manuscript. We would really appreciate it if our decision could be respected.

Reviewer 3 Report
I feel this is a clear and well written manuscript.
I suggest only a few minor changes:
- Abstract, line 34: The C is missing from F373C.
- Introduction, line 97: "Additionally, there" should be written as "Additionally, these"
- Material and methods, Clinical subjects: Lines 104 to 106. The definition of hyperurecimia is reported to be when concentration more than 420 mmol/L or 360 mmol/L depending on age and gender. I am not sure what is reported with those words. Is it Serum Uric Acid (SUA)? I think it is wort adding a sentence for clarification. Also, if it is indeed SUA, table 1 reports values in the umol/L range. Please verify values, and clarify this point.
- Results, 3.2, line 299: "No patients harbored with three"... should be "No patients harbored three"
I do not have any other comments.
Author Response
Thank you very much for your reviewing our manuscript. We are grateful that the reviewer #3 carefully checked our manuscript and acknowledged the quality of our study. Also, we apologized the typos; in the revised manuscript, all indicated-points have been amended. Additionally, as you pointed out, the definition of hyperuricemia is on the serum uric acid (μmol/L).

Round 2
Reviewer 2 Report
the authors have made some textual amendments to the paper which strengthen it. I still believe the Q141K would be a useful control in figure 4. Q141K reduces ABCG2 export of urate at the cellular level. But there is ambiguity in the literature about whether this is because it is an expression-related mutant (e.g. see Saranko et al., 2013, Kondo et al., 2004) or a transport defective mutant (e.g. see Morisaki et al., 2005, Mizuarai et al., 2004).
I don't think the paper needs this data for publication, but I do think it would strengthen the paper as it would help resolve this ambiguity in what looks like a nice experimental system.
Author Response
Thank you for your kind respect to our decision on the Q141K. A recent paper demonstrated that Q141K is, at least, an expression-related mutant with clinical relevance. In the revised manuscript, we added the relating text and a new reference.
Revised text (Line 355-358):
The reducing effect of Q141K on the ABCG2 protein level detected in the present study (Figure 3) was almost comparable to that in a previous study [40] which, like us, also used a similar transient expression strategy to validate ABCG2 mutations in vitro. This consistency supports the reliability of the following results.
New reference 40:
Zambo, B.; Bartos, Z.; Mozner, O.; Szabo, E.; Varady, G.; Poor, G.; Palinkas, M.; Andrikovics, H.; Hegedus, T.; Homolya, L., et al. Clinically relevant mutations in the ABCG2 transporter uncovered by genetic analysis linked to erythrocyte membrane protein expression. Sci Rep 2018, 8, 7487, doi:10.1038/s41598-018-25695-z.
Once again, the authors sincerely thank the reviewer #2 for his/her time and efforts of careful reading of our manuscript as well as positive feedback.